# Bailey-Bloch Congenital Myopathy in Brazilian Patients: A Very Rare Myopathy with Malignant Hyperthermia Susceptibility

**DOI:** 10.3390/brainsci13081184

**Published:** 2023-08-10

**Authors:** Gustavo Rodrigues Ferreira Gomes, Tamiris Carneiro Mariano, Vitor Lucas Lopes Braga, Erlane Marques Ribeiro, Ingred Pimentel Guimarães, Késia Sindy Alves Ferreira Pereira, Paulo Ribeiro Nóbrega, André Luiz Santos Pessoa

**Affiliations:** 1Faculty of Medicine, Ceará State University, Fortaleza 60714-903, Brazil; gustavo_r_f_gomes@hotmail.com (G.R.F.G.); ingredpguimaraes@gmail.com (I.P.G.); kesiasindy@gmail.com (K.S.A.F.P.); 2Albert Sabin Pediatric Hospital (HIAS), Fortaleza 60410-794, Brazil; dratamirismariano@gmail.com (T.C.M.); vitorlucas.vlb@gmail.com (V.L.L.B.); erlaneribeiro@yahoo.com.br (E.M.R.); 3Faculty of Medicine, Unichristus University, Fortaleza 60160-196, Brazil; paulo_r_med@yahoo.com.br; 4Division of Neurology, Department of Clinical Medicine, Federal University of Ceará, Fortaleza 60430-372, Brazil

**Keywords:** congenital myopathy, Bailey-Bloch congenital myopathy, STAC3, Native American myopathy, CMYP13, genetic disorders

## Abstract

Background: Congenital myopathy-13 (CMYP13), also known as Bailey-Bloch congenital myopathy and Native American myopathy (NAM), is a condition caused by biallelic missense pathogenic variants in *STAC3*, which encodes an important protein necessary for the excitation-relaxation coupling machinery in the muscle. Patients with biallelic pathogenic variants in *STAC3* often present with congenital weakness and arthrogryposis, cleft palate, ptosis, myopathic facies, short stature, kyphoscoliosis, and susceptibility to malignant hyperthermia provoked by anesthesia. We present two unrelated cases of Bailey-Bloch congenital myopathy descendants of non-consanguineous parents, which were investigated for delayed psychomotor development and generalized weakness. To the best of our knowledge, these are the first descriptions of CMYP13 in Brazil. In both patients, we found the previously described pathogenic missense variant p.Trp284Ser in homozygosity. Conclusion: We seek to highlight the need for screening for CMYP13 in patients expressing the typical phenotype of the disease even in the absence of Lumbee Native American ancestry, and to raise awareness to possible complications like malignant hyperthermia in Bailey-Bloch congenital myopathy.

## 1. Introduction

Congenital myopathies encompass a group of disorders with high clinical, histopathological, and genetic heterogeneity [1]. Congenital myopathy-13 (CMYP13), also known as Bailey-Bloch congenital myopathy and Native American myopathy (NAM), is a condition caused by biallelic pathogenic variants in *STAC3* located on chromosome 12q13.13-q14.1, first reported in the Lumbee Indians of North Carolina [2]. This gene encodes an important protein necessary for the excitation–relaxation coupling machinery in skeletal muscle. Its clinical presentation is mostly characterized by weakness, psychomotor development delay, short stature, kyphoscoliosis, ptosis, myopathic facies, palate abnormalities, low set ears, and congenital clubfoot. There is also risk of malignant hyperthermia among patients exposed to inhaled anesthetics or depolarizing muscle relaxants [3,4].

There are few reports of this very rare myopathy outside of Native American populations. We present two cases of Bailey-Bloch congenital myopathy in two unrelated Brazilian children born from non-consanguineous parents and without Native American ancestry.

## 2. Patients and Methods

This is a retrospective report of two patients evaluated in a specialized neurogenetics center in Brazil. Informed consent for publication was obtained for both patients. All ancillary tests were performed for the regular follow-up at the discretion of attending physicians. Genetic analysis was performed in a commercial CLIA-certified laboratory using an expanded neuromuscular genetic panel or whole exome sequencing (WES). Genetic testing (either WES or NGS panel) was performed according to availability at the center at the moment of diagnosis. Genetic analysis of the parents was not possible due to financial limitations. Variants were annotated using ANNOtate VARriation (ANNOVAR) software and Ensembl Variant Effect Predictor (VEP), and then filtered using custom R-scrips. Filtered variants were rare or absent in control population databases (gnomAD exome and gnomAD genome), and absent in homozygous individuals in controls. To detect previously reported variants, we used Pubmed in addition to ClinVar and the Leiden Open Variation Database (LOVD). The in silico predictors used were Polyphen, the Genomic Evolutionary Rate Profiling (GERP) score, and the Combined Annotation Dependent Depletion (CADD) score for missense variants, and SpliceAI for splice-site variants. Variants were classified according to the American College of Medical Genetics (ACMG) guidelines.

## 3. Results

### 3.1. Patient 1

An 8-year-old patient born from non-consanguineous parents presented a history of motor and speech delay since the age of 4 months, followed by generalized muscle weakness and feeding difficulty since the age of 3 years. He was born at full term, weighing 2.64 kg, and had infectious complications after delivery. Delayed psychomotor development was characterized by holding his head steady without support at the age of 12 months, sitting without support at 24 months, and independent gait at the age of 30 months. He had a deceased sister that was born hypotonic and passed away at the age of 4 months due to respiratory complications, two unaffected sisters, and a deceased cousin that passed away at the age of 10 years with generalized weakness.

On physical examination, the patient presented myopathic facies, facial hypomimia, bilateral ptosis, global muscle atrophy and hypotonia, areflexia, Gowers’s sign, ligament laxity, and a short stature. At the moment of the last follow-up (at the age of 8 years), he had normal speech, adequate cognitive function for his age, good social interaction, and adequate school progress (Figure 1).

Electromyography (EMG) revealed a myopathic pattern-polyphasic, short-duration, low-amplitude motor unit action potentials, and early recruitment. Creatine phosphokinase (CK) levels were normal. Pulmonary function tests (PFTs) revealed a moderate restrictive lung disease. Brain magnetic resonance imaging (MRI) was normal. Additionally, he had bilateral hearing loss (threshold of 50 dB in the right ear and 70 dB in the left), and polysomnography revealed an obstructive sleep apnea syndrome. There was important kyphoscoliosis with a Cobb Angle of 31° on spine radiography.

Whole exome sequencing revealed a homozygous pathogenic variant c.851G>C (p.Trp284Ser) in the STAC3 gene (NM_145064.3), previously reported in the Leiden Open Variation Database (LOVD: STAC3_000001) and in the literature [5,6,7]. The patient was treated with physical therapy and pulmonary rehabilitation.

### 3.2. Patient 2

A 3-year-old male child born from non-consanguineous parents presented with delayed psychomotor development, falls, and weakness. He was born hypotonic, underwent neonatal resuscitation, had prolonged respiratory insufficiency needing mechanical ventilation, and suffered from recurrent pulmonary infections. He also needed a nasogastric tube due to dysphagia. There was no family history of neurologic diseases or weakness. The patient has a healthy older brother.

Delayed psychomotor development was characterized by holding his head steady without support at the age of 6 months, sitting with support at 10 months, and walking without support at the age of 25 months.

On physical examination at the age of 3 years, the patient displayed global hypotonia, areflexia, bilateral facial paresis, myopathic and wide based gait, ogival palate, ptosis, lordosis, and bilateral congenital clubfoot (Figure 2). He was able to walk with support.

A next-generation sequencing (NGS) gene panel for neuromuscular diseases revealed the same homozygous pathogenic variant p.Trp284Ser in *STAC3.* The patient was treated with physical therapy, speech therapy for dysphagia, and pulmonary rehabilitation.

## 4. Discussion

In this study, we report the first two Brazilian cases of confirmed Bailey-Bloch congenital myopathy due to biallelic pathogenic variants in *STAC3*. Both patients are known not to have ancestry of the Lumbee Native American tribe.

The variant identified on both patients in homozygosis was p.Trp284Ser. Tryptophan at position 284 is highly conserved in different biological species and computer programs for “in silico” prediction of pathogenicity, suggesting that its replacement by serine is potentially deleterious. This variant is present in heterozygosity in 33 of approximately 141,000 individuals in a populational databank (GnomAD), and has been previously described in the medical literature, on multiple occasions, associated with Bailey-Bloch congenital myopathy [8,9]. Additionally, functional studies demonstrate that this variant leads to reduced excitation–contraction coupling in fast-twitch muscles [6]. Therefore, this variant is considered pathogenic.

The clinical phenotype was characterized by multiple congenital abnormalities in both patients, including ogival palate, congenital clubfoot, spine curvature deformities, and bilateral ptosis. Other findings were short stature, hypotonia, normal CK levels, myopathic facies, facial hypomimia, difficulties in feeding that made both patients need the use of gastric tube for feeding, and early respiratory insufficiency (Figure 1 and Figure 2). Together, these findings are in accordance with the phenotype described by Stamm et al. (2008) [10] in Native Americans with CMYP13. Additional findings such as cleft palate, arthrogryposis, low set ears, and cryptorchidism were not found in the patients in the present study. Furthermore, rarer clinical signs such as ligament laxity and facial hemangioma [11] were observed in patients 1 and 2, respectively (Table 1).

In a genetic screening research in zebrafish, Horstick et al. (2013) [6] identified *STAC3* within the genetic locus predicted in the studies by Stamm et al. (2008) [12], and attributed the mutation in this gene to the genetic basis of Bailey-Bloch congenital myopathy. STAC3 is a recently discovered essential component in the excitation–contraction coupling (EC coupling) machinery of the skeletal muscle, which is responsible for the depolarization of the membrane and consequently the rise of cytosolic calcium levels, with posterior muscle contraction. Genetic errors on the components responsible for EC coupling are associated with the development of congenital myopathies [1]. In that regard, following a cohort study with five Lumbee ancestry affected patients and thirteen non affected controls, it was established that the EC coupling defect in the CMYP13 happened due to a missense variant c.851G>C in exon 10 of *STAC3*, resulting in the substitution of tryptophan (W) for serin (S) in codon 284 on the first domain of SH3 (W284S) [6]. Ultimately, this results in the dysfunction of STAC3 in the EC coupling of the skeletal muscle due to a defective interaction between STAC3 and calcium channels and consequently a dysfunction of muscle contraction [13].

Although all individuals of the original CMYP13 cohort had homozygous missense variants, recent studies have shown compound heterozygous variants in patients without Native American ascendency [3,8,14]. In consonance with the expected pattern of mutation for CMYP13, our patients presented the biallelic pathogenic variant c.851G>C; p.Trp284Ser in *STAC3*. Table 2 shows all previously reported *STAC3* pathogenic variants and whether they were found in Lumbee or non-Lumbee patients.

Although consanguinity is very common in CMYP13 [10], and both patients had very rare variants in homozygosis, there was no confirmation of consanguinity in our patients. However, patient 1′s parents were from a small town in the hinterlands of Ceará, where consanguinity is relatively common [17], even if not explicit. Although ethnicity was self-reported, there are no reports of Native Lumbee Americans in the hinterlands of northeast Brazil, where these patients were born. Other vary rare autosomal recessive diseases have been described in this population [18,19,20]. Both families lived in Ceará for at least three generations.

An important characteristic of CMYP13 is the susceptibility for malignant hyperthermia (MH), which was described for the first time in 1987 by Bailey and Bloch in a 3-month-old Lumbee child [2]. Malignant hyperthermia is a pharmacogenetic complication with significant lethality that occurs during general anesthesia, initiated by volatile anesthetics or depolarizing muscle relaxants (succinylcholine) in susceptible patients, which results in elevation of intracellular calcium levels and, therefore, in sustained muscle contraction [21]. According to Stamm et al. (2008) [10], the mortality rate attributed to MH varies from 21% during the neonatal period to 36% in adult life. Although the risk of MH is high in CMYP13, our patients did not present any episode of MH, even when they underwent surgical procedures for correction of congenital club foot. Patient 1 performed WES while patient 2 underwent a neuromuscular disorders NGS panel. Both covered the genes *RYR1* and *CACNA1S*, and no pathogenic variants were found in those genes associated with MH.

Before the description of the first non-Native-American case of CMYP13 by Grzybowski et al. (2017) [3], this disease was considered to be exclusive to members of the Lumbee American tribe; therefore, other diagnostic hypotheses were raised for our patients due to an overlap in clinical signs including facial weakness, ptosis, hypotonia, palate alteration, and spine deformities with other entities, such as Carey–Fineman–Ziter (CFZS) and Moebius (MBS) syndromes [8]. Moebius syndrome is a congenital syndrome with the necessary presence of non-progressive facial weakness in addition to limited abduction of the eyes, associated with other congenital dysmorphisms and, rarely, MH [8,22,23,24]. Being a Brazilian patient without Native American ascendency presenting with signs of cranial nerve VI impairment, congenital anomalies, and no previous history of MH, patient 2 was initially suspected to have MBS, which was rectified only after the genetic analysis was carried out, confirming the biallelic pathogenic variant p.Trp284Ser in *STAC3*. After the diagnosis, reverse phenotyping has shown that the clinical features in our patients were very suggestive of CMYP13, as shown in Table 1.

This report raises awareness for the possibility of Bailey-Bloch congenital myopathy in patients of non-Native-American ancestry, including South American patients. Diagnostic awareness is very important to prevent complications, particularly malignant hyperthermia.

## 5. Conclusions

In short, we have described Bailey-Bloch congenital myopathy caused by the most common variant encountered in Lumbee Native Americans in two Brazilian patients not descendent of Native Americans of the Lumbee tribe. We suggest that screening for *STAC3* pathogenic variants should be considered in patients with suspicion of congenital myopathy, especially in those whose phenotypes are suggestive of CMYP13, regardless of specific Native American ancestry. In addition, early investigation of CMYP13 in these individuals is essential to avoid surgical procedures that involve anesthetics which are known to cause MH. Finally, we also raise awareness for the management of these patients, most notably in relation to the complications of this condition such as spine deformities, dysphagia, contractures, and lung capacity restriction.

## Figures and Tables

**Figure 1 brainsci-13-01184-f001:**
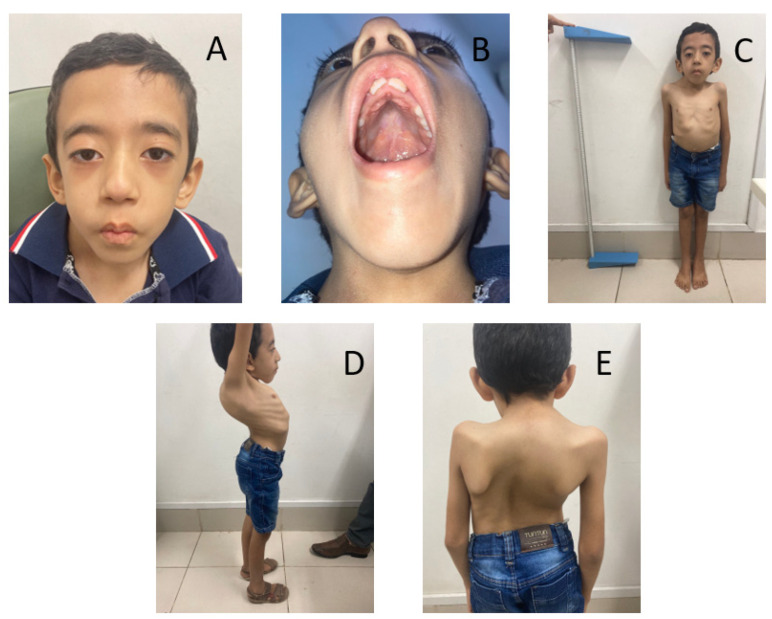
Clinical features of Bailey-Bloch congenital myopathy in Patient 1, including: (**A**) myopathic facies; (**B**) ogival palate; (**C**) short Stature; (**D**,**E**) rib cage deformities and Kyphoscoliosis.

**Figure 2 brainsci-13-01184-f002:**
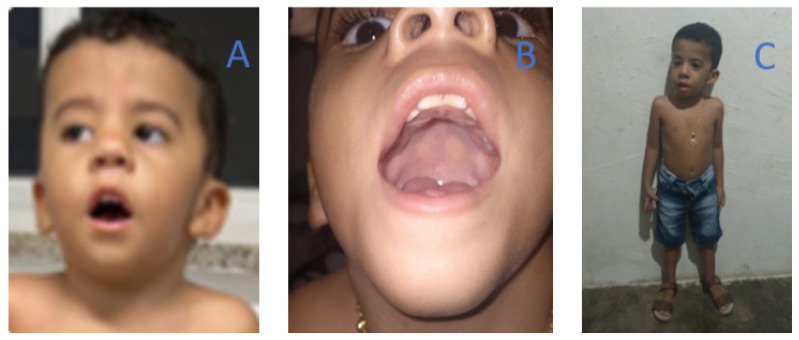
Clinical features of Bailey-Bloch congenital myopathy in Patient 2, including: (**A**) myopathic facies with low-set ears; (**B**) ogival palate; (**C**) short stature.

**Table 1 brainsci-13-01184-t001:** Clinical characteristics found in comparison to previously reported cases in Bailey-Bloch congenital myopathy and to those found in the first non-Lumbee patient reported.

	Patient 1	Patient 2	Bailey-Bloch Congenital Myopathy [11]	Grzybowski et al., 2017 [3] *
Myopathic Facies	+	+	+	+
Short stature	+	+	+	+
Congenital Club Foot	+	+	+	+
Hypotonia	+	+	+	+
Ptosis	+	+	+	+
Ogival Palate	+	+	+	+
Cleft Palate	-	-	+	-
Arthrogryposis	-	-	+	-
Cryptorchidism	-	-	+	-
Low Set Ears	+	+	+	+
Difficulty in feeding	+	+	+	+
Kyphoscoliosis	+	+	+	+
Hearing loss	+	-	+	-
Respiratory Insufficiency	+	+	+	+
Facial Hypomimia	+	+	+	+
Ligament Laxity	+	-	Rare	-
Facial Hemangioma	-	+	Rare	-

+ Present; - absent. Reference adapted from [11]. * First report of Bailey-Bloch congenital myopathy in a non-Lumbee patient.

**Table 2 brainsci-13-01184-t002:** Pathogenic variants reported in Lumbee and non-Lumbee patients.

Variant	Lumbee Ancestry	Description without Lumbee Ancestry	Authors
c.997-1G>T	No	Yes	Zaharieva et al., 2018 [14]
p.Leu255IlefsX58	No	Yes	Telegrafi et al., 2017 [8]
p.Trp284Ser	Yes	Yes	Telegrafi et al., 2017 [8] Horstick et al., 2013 [6] Zaharieva et al., 2018 [14] Gromand et al., 2022 [15]
p.Lys288Ter	No	Yes	Grzybowski et al., 2017 [3] Murtazina et al., 2022 [16]
c.432 + 4A>T	No	Yes	Grzybowski et al., 2017 [3]
p.Lys32ArgfsTer78	No	Yes	Murtazina et al., 2022 [16]

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
