# Peer review of "Bailey-Bloch Congenital Myopathy in Brazilian Patients: A Very Rare Myopathy with Malignant Hyperthermia Susceptibility"

_brainsci, 2023, doi:10.3390/brainsci13081184_

Round 1
Reviewer 1 Report
The authors present in the paper two non-related patients of brazilian origin with the very rare Native American Myopathy. The case report is well written, the photos are meaningful and the conclusions are solid.
Author Response
Dear Reviewer, we would like to thank you for the comments.
Reviewer 2 Report
The case report describes two unrelated patients with a rare variant in STAC3, leading to Bailey-Bloch congenital myopathy. Some points need to be clarified in the manuscript.
The ethnicity of the patients is not discussed in detail, and is relevant to the disease. It is stated that they are without native American ancestry. Is this self-reported or genetically confirmed? North-American studies have found that self-reported ancestry is not always reliable. Please elaborate, as the Brazilian population is interestingly heterogeneous.
Non Lumbee patients have been reported before, and I find the used abbreviation of NAM confusing, as it is also used to describe a necrotizing autoimmune myositis. Also, seeing the disorder is equally present in non-native Americans, I would like to propose an abbreviation reflecting the naming of Bailey-Bloch congenital myopathy. In addition, correct Bailey-Bloch in the title and on line 192.
Table 1 is redundant, most of the information can be found in the text. Some clarifications are needed. P2 is described as low set ears negative, however, the picture shown raises questions, please verify. Family of P2 is not described, any siblings? Relatives with weakness? How were these patients treated and disability managed, and with what success?
I suggest including a new Table, listing the STAC3 variants found so far and provide all references, of which some are indeed non-Lumbee and concern the same p.Trp284Ser.
Quality of English language is fine.
Author Response
Dear Review, thank you for the comments. Below are the responses made about the text.
The case report describes two unrelated patients with a rare variant in STAC3, leading to Bailey-Bloch congenital myopathy. Some points need to be clarified in the manuscript.
The ethnicity of the patients is not discussed in detail, and is relevant to the disease. It is stated that they are without native American ancestry. Is this self-reported or genetically confirmed? North-American studies have found that self-reported ancestry is not always reliable. Please elaborate, as the Brazilian population is interestingly heterogeneous.
Response: The reviewer raises an interesting point. Although ethnicity was self-reported, there are no reports of Native Lumbee Ameri-cans in the hinterlands of northeast Brazil, where these patients were born. Both families lived in Ceará for at least three generations. This is now stated in the discussion.
Non Lumbee patients have been reported before, and I find the used abbreviation of NAM confusing, as it is also used to describe a necrotizing autoimmune myositis. Also, seeing the disorder is equally present in non-native Americans, I would like to propose an abbreviation reflecting the naming of Bailey-Bloch congenital myopathy. In addition, correct Bailey-Bloch in the title and on line 192.
Response: We entirely agree. We changed NAM to Congenital myopathy-13 (CMYP13) and corrected the eponym Bailey-Bloch in both places.
Table 1 is redundant, most of the information can be found in the text.
Response: We believe Table 1 draws a comparison between what was found in our patients and what has been previously described in literature, highlighting the fact that Bailey-Bloch myopathy has typical findings that may suggest the diagnosis, although they are not completely homogeneous. We have added the following sentence to discussion to support the findings in the table: “After the diagnosis, reverse phenotyping has shown that the clinical features in our patients were very suggestive of CMYP13, as shown in Table 1.”
Some clarifications are needed. P2 is described as low set ears negative, however, the picture shown raises questions, please verify.
Response: We agree. The patient has low-set ears. We have corrected that. Thank you for pointing it out.
Family of P2 is not described, any siblings? Relatives with weakness? How were these patients treated and disability managed, and with what success?
Response: Patient P2 had a healthy older brother. Both patients were managed with physical therapy, speech therapy and pulmonary rehabilitation. This is now stated in the manuscript.
I suggest including a new Table, listing the STAC3 variants found so far and provide all references, of which some are indeed non-Lumbee and concern the same p.Trp284Ser.
Response: We entirely agree. We have added a table with previously published STAC3 variants and whether they have been reported in Lumbee or non-Lumbee patients.
Reviewer 3 Report
Comments:
The case report describes a known mutation p.Trp284Ser in the STAC3 gene in two unrelated patients evaluated in a specialized neurogenetics center in Brazil. This is the second case of the non amerindian NAM, after the first Turkish patient described by Grzybowski et al. (2017).
From the title it seems that the two patients described are susceptible to HM, but this was not demonstrated in the study.
Some clarifications to the authors are required:
Genetic analysis was performed in a commercial CLIA-certified laboratory. The authors could explain why two different approaches were used for genetic analysis in 2 patients: patient 1=WES, patient2=NGS.
Since the mutation is known, they could report the reference ID of the public database LOVD and the bibliographic references.
The authors should specify if family segregation has been conducted and if the parents are carriers of the same mutation p.Trp284Ser. There are other affected family members?
For comparison, the authors could add a column in the Table 1 with the clinical characteristics of the first non-Amerindian case of NAM, a Turkish 19-year-old man with compound heterozygous mutations in the STAC3 gene, described by Grzybowski et al. (2017).
Since the authors report that “According to Stamm et al. (2008) [8], the mortality rate attributed to MH varies from 21% during the neonatal period to 36% in adult life.” it could be useful to investigate the RYR1 and CACNA1S genes known to be associated with HM, to verify whether the 2 patients have a mutation in these genes
Since they did WES in patient 1, they could check for variants in these 2 genes. Are there variants in the RYR1 and CACNA1S genes in the NGS panel performed on patient 2 to verify the presence of any variant for susceptibility to HM?
This diagnostic ware very important to prevent complications of malignant hyperthermia, as the authors themselves conclude.
Small clarifications to the authors:
Line 28: the authors could insert known or described (pathogenic mutation)
Line 37: the authors could insert the
Line 90: the authors could insert the NM_145064.1 of the STAC3 gene
Lines 112-114: the authors could replace the sentence “revealed the homozygous pathogenic variant in STAC3, Chr12:57,244,322 C>G (or 112 alternatively c.851G>C - ENST00000332782), which promotes the replacement of the amino acid tryptophan by serine in codon 284 (p.Trp284Ser).” already reported for patient 1 (lines 91-92) with “revealed the same homozygous pathogenic variant p.Trp284Ser in STAC3 gene”
Lines 122-123: it is enough to report “p.Trp284Ser” in the place of “Chr12:57,244,322 C>G(or c.851G>C - ENST00000332782), which promotes the replacement of the amino acid tryptophan in codon 284 by serine (p.Trp284Ser).
Author Response
Dear Reviewer, thank you for the comments. We have addressed them and made the corrections below.
The case report describes a known mutation p.Trp284Ser in the STAC3 gene in two unrelated patients evaluated in a specialized neurogenetics center in Brazil. This is the second case of the non amerindian NAM, after the first Turkish patient described by Grzybowski et al. (2017).
From the title it seems that the two patients described are susceptible to HM, but this was not demonstrated in the study.
Some clarifications to the authors are required:
Genetic analysis was performed in a commercial CLIA-certified laboratory. The authors could explain why two different approaches were used for genetic analysis in 2 patients: patient 1=WES, patient2=NGS.
Response: Genetic testing was performed according to availability at the center at the moment of diagnosis. This is now stated in the manuscript. Thank you for this suggestion.
Since the mutation is known, they could report the reference ID of the public database LOVD and the bibliographic references.
Response: We have provided the LOVD ID with the variant description and we have added the references.
The authors should specify if family segregation has been conducted and if the parents are carriers of the same mutation p.Trp284Ser. There are other affected family members?
Response: Patient 1 had a deceased sister that was born hypotonic and passed away at the age of 4 months due to respiratory complications, two unaffected sisters and a deceased cousin that passed away at the age of 10 years with generalized weakness. Patient 2 had no affected family members. Unfortunately, segregation analysis was not performed.
For comparison, the authors could add a column in the Table 1 with the clinical characteristics of the first non-Amerindian case of NAM, a Turkish 19-year-old man with compound heterozygous mutations in the STAC3 gene, described by Grzybowski et al. (2017).
Response: We agree, and we have done that. Thank you for this suggestion.
Since the authors report that “According to Stamm et al. (2008) [8], the mortality rate attributed to MH varies from 21% during the neonatal period to 36% in adult life.” it could be useful to investigate the RYR1 and CACNA1S genes known to be associated with HM, to verify whether the 2 patients have a mutation in these genes. Since they did WES in patient 1, they could check for variants in these 2 genes. Are there variants in the RYR1 and CACNA1S genes in the NGS panel performed on patient 2 to verify the presence of any variant for susceptibility to HM?
Response: We agree. Patient 1 performed WES while patient 2 underwent a neuromuscular disorders NGS panel. Both covered the genes RYR1 and CACNA1S, and no pathogenic variants were found in those genes associated with MH. This is now stated in the discussion.
This diagnostic ware very important to prevent complications of malignant hyperthermia, as the authors themselves conclude.
Small clarifications to the authors:
Line 28: the authors could insert known or described (pathogenic mutation)
Response: We agree. We have corrected that.
Line 90: the authors could insert the NM_145064.1 of the STAC3 gene
Response: We agree. We have changed that according to the reviewer´s suggestion.
Lines 112-114: the authors could replace the sentence “revealed the homozygous pathogenic variant in STAC3, Chr12:57,244,322 C>G (or 112 alternatively c.851G>C - ENST00000332782), which promotes the replacement of the amino acid tryptophan by serine in codon 284 (p.Trp284Ser).” already reported for patient 1 (lines 91-92) with “revealed the same homozygous pathogenic variant p.Trp284Ser in STAC3 gene”
Response: We agree. We have corrected that.
Lines 122-123: it is enough to report “p.Trp284Ser” in the place of “Chr12:57,244,322 C>G(or c.851G>C - ENST00000332782), which promotes the replacement of the amino acid tryptophan in codon 284 by serine (p.Trp284Ser).
Response: We agree. We have corrected that.
Reviewer 4 Report
The case report describes the case of 2 patients with Bailey-Bloch disease outside native american ancestry. The case study is well written and covers the main aspects of genetics and disease description. I have no comments for further improvement
English language quality is sound. Some minor spelling errors should be checked carefully bevor final submission.
Author Response
Dear reviewer, thank you for the comments made.
Round 2
Reviewer 2 Report
The manuscript has been properly revised, improving its quality to publication ready.
Author Response
Dear Reviewer, thank you for the comments and for the suggestions made.
Reviewer 3 Report
Some clarifications to the authors are required:
1. Genetic analysis was performed in a commercial CLIA-certified laboratory. The authors could explain why two different approaches were used for genetic analysis in 2 patients: patient 1=WES, patient2=NGS. - ok
2. Since the mutation is known, they could report the reference ID of the public database LOVD and the bibliographic references. - change as follows
3. The authors should specify if family segregation has been conducted and if the parents are carriers of the same mutation p.Trp284Ser. There are other affected family members? - The authors clarified that no family member is affected (parents and sibling), but they should specify why the parents were not analyzed for the same mutation reported by the child.
4. For comparison, the authors could add a column in the Table 1 with the clinical characteristics of the first non-Amerindian case of NAM, a Turkish 19-year-old man with compound heterozygous mutations in the STAC3 gene, described by Grzybowski et al. (2017). - ok
5. Since the authors report that “According to Stamm et al. (2008) [8], the mortality rate attributed to MH varies from 21% during the neonatal period to 36% in adult life.” it could be useful to investigate the RYR1 and CACNA1S genes known to be associated with HM, to verify whether the 2 patients have a mutation in these genes. Since they did WES in patient 1, they could check for variants in these 2 genes. Are there variants in the RYR1 and CACNA1S genes in the NGS panel performed on patient 2 to verify the presence of any variant for susceptibility to HM? This diagnostic ware very important to prevent complications of malignant hyperthermia, as the authors themselves conclude. - ok
Small clarifications to the authors:
Line 28: the authors could insert known or described (pathogenic mutation), ok
Line 90: the authors could insert the NM_145064.1 of the STAC3 gene, change the whole sentence with: Whole exome sequencing revealed a homozygous pathogenic variant c.851G>C (p.Trp284Ser) in the STAC3 gene (NM_145064.3) previously reported in Leiden Open Variation Database (LOVD: STAC3_000001) and in literature.
Line 97: enter 3 references
doi: 10.1038/ncomms2952.
doi: 10.1038/ejhg.2016.146.
doi: 10.1038/s41598-022-22036-z.
then check reference list
Lines 112-114: the authors could replace the sentence “revealed the homozygous pathogenic variant in STAC3, Chr12:57,244,322 C>G (or 112 alternatively c.851G>C - ENST00000332782), which promotes the replacement of the amino acid tryptophan by serine in codon 284 (p.Trp284Ser).” already reported for patient 1 (lines 91-92) with “revealed the same homozygous pathogenic variant p.Trp284Ser in STAC3 gene”- ok
Lines 122-123: it is enough to report “p.Trp284Ser” in the place of “Chr12:57,244,322 C>G
(or c.851G>C - ENST00000332782), which promotes the replacement of the amino acid tryptophan in codon 284 by serine (p.Trp284Ser).- ok
Table 2: replace the mutation p.Leu255fs with p.Leu255IlefsX58.

Author Response
Dear reviewer, All pending queries were addressed in this last version. Thank you for the comments and suggestions made